# Is Neonatal Uterine Bleeding Involved in Early-Onset Endometriosis?

**DOI:** 10.3390/biom14050549

**Published:** 2024-05-03

**Authors:** Marwan Habiba, Sun-Wei Guo, Giuseppe Benagiano

**Affiliations:** 1Department of Health Sciences, University of Leicester and University Hospitals of Leicester, Leicester LE1 5WW, UK; 2Department of Biochemistry and Molecular Biology, Research Institute, Shanghai Obstetrics and Gynecology Hospital, Fudan University, Shanghai 200011, China; hoxa10@outlook.com; 3Faculty of Medicine and Surgery, “Sapienza” University of Rome, 00161 Rome, Italy; giuseppe.benagiano@uniroma1.it; 4Geneva Foundation for Medical Education and Research, 1202 Geneva, Switzerland

**Keywords:** endometriosis, neonatal uterine bleeding, early-onset endometriosis, retrograde menstruation, endometrium

## Abstract

Background: There has been considerable progress in our understanding of endometriosis, but its pathophysiology remains uncertain. Uncovering the underlying mechanism of the rare instances of endometriosis reported in early postmenarcheal years and in girls before menarche can have wide implications. Methods: We conducted a literature review of all relevant articles on Medline. Results: In the review, we explore the pathogenetic theories of premenarcheal endometriosis, the role of retrograde menstruation in the adult and its potential role in early-onset disease, as well as the factors that argue against the existence of a link between early-onset endometriosis (EOE) and neonatal uterine bleeding (NUB). Conclusions: As with endometriosis in adult women, the pathogenesis of early-onset disease remains unclear. A link between NUB and EOE is plausible, but there are considerable challenges to collating supporting evidence. The state of our understanding of early uterine development and of the pathophysiology of NUB leaves many unknowns that need exploration. These include proof of the existence of viable endometrial cells or endometrial mesenchymal stem cells in NUB, their passage to the pelvic cavity, their possible response to steroids, and whether they can reside within the pelvic cavity and remain dormant till menarche.

## 1. Introduction

A number of pathogenetic pathways and hypotheses have been proposed to explain the development of endometriosis. These include retrograde menstruation, coelomic metaplasia, benign metastasis, immune dysregulation, hormonal imbalance, involvement of stem cells and alterations in epigenetic regulation [1,2,3]. We initiated a series of critical reviews of the major theories on the pathogenesis of endometriosis, pointing to strengths, weaknesses, and uncertainties in these hypotheses, including those mentioned in our previous work [4,5,6]. In this review, we address the question of the origin of early-onset endometriosis.

Endometriosis is most commonly identified in reproductive-aged women. It is rare during the early postmenarcheal years and only a few cases have been reported in girls before menarche. The latter group is particularly interesting because they challenge the commonly held view of the association between endometriosis and retrograde menstruation. The term early-onset endometriosis (EOE) is used here to refer to cases of endometriosis in premenarcheal and early postmenarcheal (adolescent) girls.

The identification of cases with EOE raises the question of whether this variant has the same origin as the adult disease. Goldstein et al. [7] opined that endometriosis detected a few months after menarche may be congenital and that the onset of ovulatory cycles in such cases leads to the stimulation of inactive endometrial tissue in the pelvic peritoneum or to the differentiation of totipotent cells. It was also hypothesized that EOE may be rooted in neonatal development and linked to the phenomenon of neonatal uterine bleeding (NUB) [8,9]. It was postulated that NUB can be associated with retrograde flow into the peritoneal cavity resulting in the seeding of mesenchymal stem cells (MSCs) with the ability to proliferate and differentiate and lead to later development of endometriosis [10]. Endometrial stem cells are known to be present both in the basalis and the functionalis layers of the endometrium and may have different characteristics in women with and without endometriosis [11,12,13].

In this review, we focus on the evidence in favor of and against the neonatal origin of at least some cases of perimenarcheal endometriosis.

## 2. Material and Methods

We undertook a Medline search using the terms “premenarcheal endometriosis”, or “early-onset endometriosis” (n = 12), “neonatal uterine bleeding”, or “neonatal menstruation” (n = 18), and neonate or newborn plus uterine bleeding or hemorrhage plus endometriosis (n = 23). Identified articles were combined (n = 37) and searched manually to select all articles that addressed the relationship between NUB and endometriosis (n = 26). The latter were reviewed in full text to extract relevant information. We also undertook a search using the terms “adolescent” or “pre-menarche” or “menarche” (n = 2,279,914) and “endometriosis” (n = 34,073). These were combined yielding 2264 articles, which in turn were combined with the search terms “pathophysiology” and “etiology or cause”, yielding 196 articles that were searched based on title and abstract, and 21 were selected for full-text review.

## 3. Evidence of the Existence of Early-Onset Endometriosis

The first description of premenarcheal endometriosis is perhaps that reported in *JAMA* in 1948 of a girl who at the age of 11 years and 8 months, complained of monthly abdominal pain that started 8 months before menarche and became increasingly severe after menarche. At surgery, a mass attached to the left cornu of a bicornuate uterus contained numerous endometrial cysts filled with a chocolate-colored substance [14]. The authors considered this a case of endometriosis. Some 65 years later, Gogacz et al. [15] described a case in a prepubertal girl with an ovarian endometrioma with hemosiderin-loaded macrophages lining the ovarian cortex.

Several reports of endometriosis in adolescence were published in the 1960s. An article by Derryberry and Bonney [16] described the case of a 15-year-old girl with peritoneal endometriosis. This was followed by a small series by Hanton et al. [17]. Additional cases and small series followed [18,19,20]. In 1980, Goldstein et al. reported a case that was 5 months postmenarche [7]. Of relevance is the occurrence of endometriosis in adolescents with uterine anomalies causing menstrual outflow obstruction [21].

Reese et al. [22] examined 67 adolescent girls who had not responded to analgesia or oral contraceptives administered for pelvic pain. At laparoscopy, endometriosis (in the vast majority made up of superficial red stage 1 lesions) was present in 49 (73%) of cases. Marsh and Laufer [23] published a series of five premenarcheal girls aged 8.5–13 years with Tanner stage I–III breast development who had had pelvic pain lasting for more than 6 months and normal pelvic ultrasound. Laparoscopy identified clear, red, and/or white peritoneal lesions that featured vascular proliferation, hemosiderin deposits, macrophage proliferation, and stroma, but no endometrial glands, inflammation, granulation, endosalpingiosis, or adhesions.

Recently, Rasp et al. [24] identified 526 cases of adolescent girls (aged from <17 to 19 years) from a health register from Finland over a period of 26 years who had had a surgical diagnosis of endometriosis. Most cases (72%) had superficial peritoneal and 23% ovarian endometriosis. No breakdown was provided of the time since menarche, but only a quarter of the cases were <15 years old.

## 4. Classic Pathogenetic Theories of Perimenarcheal Endometriosis

Marsh and Laufer [23] described five cases of premenarcheal endometriosis. They favored the possibility that the disease originates from coelomic metaplasia or from Müllerian embryonic rests, rather than from retrograde menstruation, as proposed in Sampson’s theory [25]. Coelomic metaplasia and Müllerian embryonic rests theories were initially proposed at the beginning of the 20th century [26], and were subsequently considered alternatives to Sampson’s theory or as having a place in relation to particular subtypes of endometriosis. Briefly, the coelomic metaplasia theory, first proposed in 1923 and 1924 by Meyer [27,28] and expanded on by Gruenwald in 1942 [29], theorizes that endometriosis is formed from the peritoneum or pleura and that it is therefore of non-Müllerian origin. The Müllerian embryonic remnants theory postulates that during fetal organogenesis, fragments of the coelomic epithelium—which is the precursor of the Müllerian duct epithelium and therefore of the endometrium—are seeded in the peritoneal cavity where they lie dormant until menarche, when they transform into endometrial cells as a response to exposure to endogenous hormones [30].

Batt and Mittwally [31] took the view that EOE develops from embryonic Müllerian rests and proposed that thelarche, rather than chronological age, be recognized as the reference point, since it marks the onset of estrogen stimulation of preexisting embryonic Müllerian rest–endometriosis. This suggest that the disease can become symptomatic before menarche [20].

## 5. The Novel Pathogenetic Hypothesis Involving Neonatal Uterine Bleeding

Some 10 years ago, Brosens and his group [8,9] put forward the hypothesis that NUB, i.e., bleeding during the neonatal period, may be implicated in the pathogenesis of EOE. They stressed that this hypothesis can be challenging to prove because of the time lag between NUB and the identification of endometriosis. NUB is also often overlooked as being a physiological phenomenon of no consequence, and as such is rarely recorded. It is difficult to prove that retrograde bleeding occurs in cases of NUB, and even if peritoneal reflux were demonstrated, there would remain a need to examine its cellular content, which can differ from the components expelled through the vagina.

We recently reviewed the possible pathogenetic mechanisms involved in NUB [32] and concluded that the response of the fetal endometrium to hormonal challenges differs from that of the adult, since progestogenic response (i.e., secretory or decidual changes) is seen in only a minority of cases [33]. Fetal endometrial development may be affected by a number of factors, including intrauterine stressors and gestational age, and endometrial response to hormone withdrawal at birth can therefore vary.

### 5.1. Retrograde Menstrual Bleeding in the Adult and the Neonate

Although initially met with skepticism, Sampson’s theory of retrograde menstruation [25] as the principal mechanism for peritoneal endometriosis is now widely recognized. Some earlier researchers favored the serosal or celomic metaplasia hypothesis. As stressed by Yovich et al. [34], Novak [35], and others, there are a series of objections to the retrograde menstruation hypothesis. These include the beliefs that:(i)Menstrual blood rarely, if ever, escapes from the uterine cavity into the tubes.(ii)The lumen of the interstitial portion of the tube is too small to allow the retrograde passage of endometrial tissue fragments.(iii)Endometrial tissue set free by menstruation is dead or dying and therefore is unable to implant and grow.(iv)Several days are required for endometrial tissue to be carried from the uterine cavity through the tubes, which leaves little chance for such “degenerating tissue” to grow where it falls.

Sampson countered most of these objections and demonstrated that menstrual blood can reach the peritoneal cavity first and foremost as a backflow, through the tubes, from the uterine cavity, and from the menstrual reaction of the tubal mucosa. Today, the occurrence of retrograde menstruation in adult women is a recognized occurrence. However, there is disagreement on its frequency, recurrence, and cellular content. An early investigation by Polishuk and Sharf using culdoscopy reported retrograde menstruation in 50% of cases [36]. Subsequently, Halme et al. [37] utilized laparoscopy performed during the perimenstrual phase and documented the presence of blood in the peritoneal fluid (PF) in 90% of women with patent tubes. They also observed evidence of blood in the peritoneum in 15% of cases where the fallopian tubes were occluded. The same phenomenon with the same frequency was documented in nine patients with endometriosis. In 2009, Bokor et al. [38] obtained PF at laparoscopy from 28 women during the menstrual phase and compared this with samples obtained from 84 women during other phases of the cycle. They found that the concentration of red and white blood cells in the PF is increased during menstruation. Immunochemically detected endometrial cells were low, even during the menstrual phase [38]. A similar observation was made back in 1986 by Bartosik et al. [39], who searched for the presence of endometrial tissue in the PF in a series of 67 women with documented tubal patency. Endometrial tissue was found in only a small proportion of women. Interestingly, there was no significant difference between patients with and without endometriosis (19% vs. 11%; *p* = 0.6) [39].

The recent review by Vercellini et al. [36] concluded that there is variability in the reported incidence and inconsistency in finding of endometrial cells or tissue in the peritoneal cavity. Consequently, they concluded that the role of retrograde menstruation should not be taken for granted, as it has not been conclusively proven. This has implications when considering the pathogenesis of EOE [40]. Another important but poorly understood question is the duration of time needed from first exposure to retrograde menstruation to disease development. This may be distinct from the question of repeat exposure, but as the interval is not known, the distinction between EOE and adolescent disease becomes effectively blurred.

Little is known about retrograde menstruation around menarche. A small study by Turner and Coulthard [41] followed three girls who reached menarche while undergoing peritoneal dialysis and observed cyclical blood staining of peritoneal dialysis fluid in all cases. It is plausible but uncertain if bleeding originated from retrograde menstruation. Interestingly, this phenomenon preceded the occurrence of vaginal bleeding. Whether retrograde menstruation occurs in preference to vaginal bleeding is unknown. It is important to note that bleeding has been found during peritoneal dialysis in cases where it was not related to menstruation, including cases reported in men [42]. Kruitwagen et al. [43] cultured cells from the PF during the early follicular phase (days 1–7) in 24 women and identified endometrial epithelial cell colonies in 19 samples. There was a wide variation (1 to 200 or more per PF sample), although no significant difference was found between women with minimal endometriosis (n = 11) and those unaffected (n = 12). Endometrial epithelial cell colonies were identified in culture in 19 of 24 women [43]. Bokor et al. [34] questioned the nature of these cells. There was no significant difference in the incidence of cells in the PF in women with endometriosis (67%) and those without endometriosis (92%) [44]. Similar observations were reported by Bartosik et al. [39]. However, Al-Badawy et al. [45] identified endometrial tissue more frequently in women with endometriosis.

It remains possible but unproven that NUB is associated with retrograde flow in newborns, and the chances may be enhanced by the anatomical structure of the hypertrophied neonatal cervix. However, it is acknowledged that the opportunity to confirm this phenomenon is limited, as it requires the incidental need to examine the peritoneal cavity of a neonate during a narrow window of a few days.

### 5.2. Composition of the Menstrual Blood in Adults

Almost 60 years ago, Burnhill et al. [46] analyzed the content of menstrual blood obtained from 260 women. Thirty percent of the samples contained tissue on all occasions, 35% were persistently negative, and in a large number of samples, there was insufficient tissue for diagnostic purposes. They concluded that “[T]he generally held opinion that tissue is shed in the early days of the menstrual cycle is confirmed. It is also shown that tissue can be shed in anovulatory cycles” [46].

More recent studies, incorporating modern technologies, enhanced our understanding of the composition of menstrual fluid. Menstrual fluid contains a variety of immune cells [47,48] and endometrial mesenchymal stem cells [49]. It also contains viable endometrial mesenchymal stem cells (eMSCs) and endometrial epithelial progenitor cells (eEPCs), which are capable of survival to form colonies. The intra- and interindividual variation in cellular and protein content of menstrual fluid is small, and the cellular composition of menstrual fluid is likely to be representative of the eutopic endometrium [50]. There may be differences in cellular content between uterine menstrual blood recovered from the uterus and that obtained from the vagina, and it is also possible that not all eutopic endometrial cells are shed equally or that they have the same rate of viability [50]. It has also been proposed that variations may exist between the cellular composition of menstrual fluid in cases of irregular shedding, which may increase the chance of endometriosis [51].

### 5.3. Composition of Uterine Blood in Neonates

In a recent study, Ogawa et al. [52] set out to evaluate the hypothesis that NUB is responsible for EOE. The study examined neonatal endometria (n = 15) obtained postmortem and blood collected from diapers from neonates with NUB (n = 18). The blood was collected with tweezers or by washing soiled diapers in Dulbecco’s modified essential medium. Samples were examined for biological markers of endometrial mesenchymal stem cells (eMSCs) CD90/CD105, steroid receptors (ER/PGR), decidualization (prolactin, IGFBP1), pre-decidualization (glycodelin A, α-SMA), proliferation (Ki-67 index), vascularity (CD31+ cells), and immunocompetent CD68+, CD45+, and CD56+ cells. Cell transfer and immunocytochemistry were utilized to examine eMSCs and/or collected endometrial cells. No endometrial cells or eMSCs were identified in samples (n = 18). This led them to conclude that there was no link between NUB and early-onset endometriosis. Postmortem neonatal uterine endometrium (n = 15) exhibited variable expression of some phenotypes of eMSCs (CD90/CD105).

It is well documented that the adult endometrium contains mesenchymal stem/stromal cells (MSCs). It has been proposed that the term “menstrual blood stem/stromal cells” (MenSCs) be used to distinguish cells in menstrual fluid from endometrial MSCs (eMSCs) in the endometrium [49]. Standardization of MenSC isolation protocols and culture conditions are challenging, but most have relied on collection of a volume of menstrual blood using a cup device. The technique adopted by Ogawa et al. [52] relied on the extraction of blood and menstrual debris from diapers, and can be affected by factors such as time, contamination, cell adhesion to the diapers, and cell viability. The tiny volume of biological material obtained was then subjected to pelleting, cell transfer, further fractionation, and processing through immunohistochemistry without antigen retrieval. Thus, the failure to identify MenSC may be related to methodological constraints.

Study of postmortem uteri has long been hampered by the difficulty of obtaining suitable tissue for study. Ogawa et al. [52] collected 15 samples from fetuses/neonates ranging between 27–40 weeks’ gestation who died 0–32 days after birth. The timing from death to tissue harvesting is critical to the quality of the material used for the study, but routine autopsy can take several hours, if not days, resulting in autolysis. This delay may explain the variable expression reported here.

The absence of cellular debris in vaginal blood has been interpreted by Ogawa et al. [48] as proof that NUB cannot be involved in EOE, as proposed by Brosens et al. [8,9,53] and Gargett et al. [10]. However, several points need clarification. First, in enunciating their theory, Brosens et al. [9] pointed out that by mid-gestation the fetal uterus has a “defined lumen”, as proven by Terruhn [54]. Of importance, the cervical canal is not patent after 26 weeks’ gestation because of a cervical plug. Also, the cervix of a newborn girl is between 2 and 2.5 times that of the uterine corpus [55]. The long cervix may add resistance to menstrual efflux and/or create selective filtering. Whether this results in retention of endometrial debris remains speculative. Importantly, over a century ago, Halban [56] observed the presence of endometrial breakdown and microscopic blood extravasation in 8 out of 21 neonatal uteri examined at autopsy. Given the observation by Ogawa et al. [52], endometrial debris remain to be accounted for. Proving or disproving the hypothesis that NUB has a role in the genesis of EOE requires the presence, rare as it may be, of endometrial shedding after birth, i.e., that the endometrium refluxes, implants, and remains dormant in the peritoneal cavity till stimulated. This is why Brosens et al. [8,9] emphasized the importance of factors that increase the likelihood of peritoneal reflux of blood.

The reported incidence of endometriosis in teenagers ranges between 11% and 40% [57]. The risk of adolescent endometriosis increases in the presence of Müllerian anomalies associated with outflow tract obstruction [21,57]. Thus, the starting point is whether peritoneal reflux occurs (and its content) in neonates with NUB. At present, there is no direct information on this vital point.

Another reason for caution in relation to the conclusions drawn by Ogawa et al. [52] is the finding of variable expression of some phenotypes of eMSCs. The presence of stem cells in neonatal endometrium supports the hypothesis first put forward by Gargett et al. [10] “that stem/progenitor cells present in shedding endometrium may have a role in the pathogenesis of early-onset endometriosis through retrograde neonatal uterine bleeding”. As shown in postmenopausal women, these cells can remain dormant for many years [58]. This lends plausibility to the hypothesis that in the presence of retrograde neonatal menstrual bleeding, progenitor cells seed in the pelvis soon after birth. These cells can remain quiescent until stimulated by the rising estrogen preceding menarche. Estrogen stimulates angiogenesis, cellular proliferation, and invasiveness, promoting the formation of endometriotic lesions [9]. Another challenge for the study of Ogawa et al. [52] is the moderate sample size. For n = 18, the eMSC-positivity rate has to be at least 0.25% to have a probability of 5% when at least one sample turns positive for eMSCs.

### 5.4. Clinical Evidence

In a recent retrospective clinical study, Tandashvili et al. [59], reported a 2.4% incidence of NUB. The incidence of NUB (as reported by their mothers) was 1.8% (n = 9) in 500 women with endometriosis compared to 0.3% (n = 1) in 350 patients without endometriosis. NUB occurred in babies of 55.6% of women who themselves had NUB. There were no cases of NUB amongst babies born to women who did not have NUB themselves (*p* < 0.0001). Although the question of recall bias is difficult to resolve, this retrospective study suggests a link between NUB and endometriosis later in life. However, a prospective web-based survey of adult women (20–26 years old) who as neonates either had (responders, n = 31; contracted, n = 105) or did not have (responders, n = 52; contacted, n = 149) NUB did confirm an association between NUB and the occurrence of symptoms related to endometriosis [59].

The more recent retrospective study by Ogawa et al. [60] examined contemporary and historical medical records to assess the association between NUB and endometriosis-related symptoms in young women. The study included two separate cohorts. In the first cohort of 1093 female neonates born between 1996 and 2000, NUB occurred in 9.6% of cases. In the second cohort of 807 female neonates born between 2013 and 2017, the incidence of NUB was only 3.1%. While recall bias could not be ruled out, the study suggested that newborns with a history of NUB were more likely to complain of endometriosis-related symptoms during adulthood.

## 6. Countering Factors

Whether NUB is associated with retrograde bleeding remains an open question. However, as is the case in adults, even if proven, the occurrence of retrograde menstruation per se is not sufficient proof of a relationship to endometriosis. The difficulty is further compounded because NUB presumably occurs only once, and over only a few days. Therefore, the window of opportunity for viable endometrial cells to travel to the peritoneal cavity is limited to 3–5 days. Nothing is known about the potential for neonatal endometrial cells to survive or implant. How that may be affected by the prevailing hormonal milieu is also unknown. The adult endometrium contains at least two stem/progenitor cell populations: the epithelial progenitor cells (eEPCs) and the mesenchymal stem cells (eMSCs). These cells are capable of differentiation to form new endometrial tissue. Endometrial MSCs are not confined to the basalis, but are also present in the functionalis and thus in the shed endometrium [10]. The question remains about the quantity of eMSC required for implantation and disease genesis. The quantity of endometrium shed during adult menstruation is known to vary, but there is no research on the volume of endometrium shed at the time of NUB. The size of the neonatal uterus and the endometrium therein is small. Animal studies have shown that successful development of endometriosis increases in proportion to the volume of endometrium deposited into the peritoneum. It remains possible that the content of retrograde bleeding is qualitatively different from that recovered vaginally. Curiously, this opens the possibility that retrograde bleeding (whether linked to EOE) can occur in the absence of manifest vaginal loss. It is important here to consider that the incidence of NUB is higher if tested for biochemically [32].

The presence of a cervical plug in the female neonate has been proven [54], but little is known about the fallopian tubes. In addition, nothing is known about the factors that may influence neonatal (or for that matter, even adult) tubal reflux, e.g., whether that is affected by the recumbent position of the neonate or by oxytocin.

As discussed, whether retrograde bleeding occurs in the neonate remains to be proven and the cellular content (if any) that may be carried into the peritoneum is speculative. However, it is recognized that neonatal uterine development is not complete at birth and that the endometrium at birth exhibits a spectrum of hormone response, rather than being uniformly secretory. This suggests that the neonatal endometrium may harbor cells capable of further differentiation and maturation and that its response to steroids, including the occurrence of NUB, can be an indicator of preceding events and of future potential. This leaves open the possibility that NUB is linked to endometriosis through either a link to Sampson’s hypothesis or alternative mechanisms linked to differential development.

Last, but not least, it is uncertain of the mere presence of viable endometrial cells is sufficient to establish endometriotic foci. Leyendecker et al. reported a significantly higher prevalence of fragments of shed basalis in menstrual fluid of women with endometriosis compared to unaffected women [61], yet endometrial basalis is located at the endometrial–myometrial interface (EMI) or the junctional zone (JZ). This region is known to be the origin of uterine contraction [62], the region where endometrial stem cells were first identified [63,64]. It is also known to be densely innervated with peripheral nerves, maintained by unsheathing Schwann cells (SCs) that are derived from the neural crest [65]. SCs are the most important type of glial cells in the peripheral nerve system and are distributed nearly ubiquitously in the human body.

## 7. Conclusions

As with endometriosis in adult women, the pathogenesis of EOE remains unclear. The issue is complicated because the age cutoff for distinguishing EOE from adolescent disease is unclear. Delayed diagnosis renders it impossible to pinpoint the precise age of onset. As alluded to above, a link between NUB and EOE is plausible, but there are considerable challenges to collating supporting evidence. The state of our understanding of early uterine development and of the pathophysiology of NUB leaves many unknowns that need exploration. This includes the proof of the existence of viable endometrial cells or eMSCs in NUB, their passage to the pelvic cavity, their response to steroids, and whether they are able to reside within the pelvic cavity and remain dormant till menarche. Any endeavor to investigate these is challenging. It is also unknown if retrograde bleeding occurs in the absence of manifest loss, and the significance of the observed differences in neonatal endometrial features on future uterine development or disease remain to be examined.

## Data Availability

Not applicable.

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
