# Peer review of "Is Neonatal Uterine Bleeding Involved in Early-Onset Endometriosis?"

_biomolecules, 2024, doi:10.3390/biom14050549_

Round 1
Reviewer 1 Report
Comments and Suggestions for Authors
Abstract is not good enough. Should be written in perfect native english and more informative.
Lines 54-61- it seems an excessive description of a clinical case from 1948 that may have some insifficiencies in light of current knowledge
INTRODUCTION should be written in a different way. Authors emphasize clinical cases instead of describing more recent studies and series with scientific evidence. Authors also should provide accurate information about pathophysiology theories of pre-manarcheal endometriosis, namely mesenchymal stem cell theory that seems of major relevance in order to explain this subject. For instance, recent references as
Liu Y., Zhang Z., Yang F., Wang H., Liang S., Wang H., Yang J., Lin J. The role of endometrial stem cells in the pathogenesis of endometriosis and their application to its early diagnosis. Biol. Reprod. 2020;102:1153–1159. doi: 10.1093/biolre/ioaa011.
Cordeiro, M.R.; Carvalhos, C.A.; Figueiredo-Dias, M. The Emerging Role of Menstrual-Blood-Derived Stem Cells in Endometriosis. Biomedicines 2023, 11, 39. https://doi.org/10.3390/biomedicines11010039
Comments on the Quality of English LanguageExtensive editing of English required.
Author Response
Reviewer 1
Abstract is not good enough. Should be written in perfect native english and more informative.
Response: We have revised the Abstract as suggested
Lines 54-61- it seems an excessive description of a clinical case from 1948 that may have some insifficiencies in light of current knowledge
Response: We have trimmed down the description in recognition of the point made by the referee.
INTRODUCTION should be written in a different way. Authors emphasize clinical cases instead of describing more recent studies and series with scientific evidence. Authors also should provide accurate information about pathophysiology theories of pre-manarcheal endometriosis, namely mesenchymal stem cell theory that seems of major relevance in order to explain this subject. For instance, recent references as
Liu Y., Zhang Z., Yang F., Wang H., Liang S., Wang H., Yang J., Lin J. The role of endometrial stem cells in the pathogenesis of endometriosis and their application to its early diagnosis. Biol. Reprod. 2020;102:1153–1159. doi: 10.1093/biolre/ioaa011.
Cordeiro, M.R.; Carvalhos, C.A.; Figueiredo-Dias, M. The Emerging Role of Menstrual-Blood-Derived Stem Cells in Endometriosis. Biomedicines 2023, 11, 39. https://doi.org/10.3390/biomedicines11010039
Response: We have restructured the introduction as suggested and removed the section containing the clinical cases from the introduction. We have added the refernces suggested and also added:
Penariol, L.B.C.; Thomé, C.H.; Tozetti, P.A.; Paier, C.R.K.; Buono, F.O.; Peronni, K.C.; Orellana, M.D.; Covas, D.T.; Moraes, M.E.; Silva,W.A., Jr.; et al. What Do the Transcriptome and Proteome of Menstrual Blood-Derived Mesenchymal Stem Cells Tell Usabout Endometriosis? Int. J. Mol. Sci. 2022, 23, 11515.
We have undertaken a thorough language review and corrected where relevant.
Reviewer 2 Report
Comments and Suggestions for Authors
To the Authors
First of all, let me to send you the:best compliments for the review you realized about an important topic of Gynecology, that one of the Authors (G.B. ) is denoting scientific interest from a long time ( ten years about ).I have already envoyed to Editors, my decision as requested, And by writing now to you I would likei to let you know about my comments on your submitted paper .
My opinion on the submitted review as presented is that denotes without any doubt scientifically sound and is conducted with great knowledge as well as high criticism regarding the problems the Authors consider and discuss to. The references are appropriate and English language is fine..
I am fully convinced that this submitted review idserves an high consideration for to be directly published in Biomolecules in the current form,without any revision.
Kind regards
Author Response
Reviewer 2
First of all, let me to send you the:best compliments for the review you realized about an important topic of Gynecology, that one of the Authors (G.B. ) is denoting scientific interest from a long time ( ten years about ).I have already envoyed to Editors, my decision as requested, And by writing now to you I would likei to let you know about my comments on your submitted paper .
My opinion on the submitted review as presented is that denotes without any doubt scientifically sound and is conducted with great knowledge as well as high criticism regarding the problems the Authors consider and discuss to. The references are appropriate and English language is fine..
I am fully convinced that this submitted review idserves an high consideration for to be directly published in Biomolecules in the current form,without any revision.
Response: Thank you very much for your review
Round 2
Reviewer 1 Report
Comments and Suggestions for Authors
The revised version provided by the authors has been sufficiently improved to warrant publication in Biomolecules.